# Lipid-Lowering Therapy in Patients with Coronary Heart Disease and Prior Stroke: Mission Impossible?

**DOI:** 10.3390/jcm10040886

**Published:** 2021-02-22

**Authors:** Pier Luigi Temporelli, Marcello Arca, Laura D’Erasmo, Raffaele De Caterina

**Affiliations:** 1Division of Cardiology, Istituti Clinici Scientifici Maugeri, IRCCS, Via Revislate 13, 28013 Gattico-Veruno, Italy; 2Department of Translational and Precision Medicine, Sapienza University of Rome, Viale dell’Università 37, 00161 Roma, Italy; marcello.arca@uniroma1.it (M.A.); laura.derasmo@uniroma1.it (L.D.); 3Chair of Cardiology, Cardiovascular Division, Pisa University Hospital, University of Pisa, Via Paradisa 2, 56126 Pisa, Italy; raffaele.decaterina@unipi.it; 4Fondazione Villa Serena per la Ricerca, 65013 Città Sant’Angelo, Italy

**Keywords:** coronary heart disease, stroke, lipid-lowering therapy, cholesterol, triglycerides

## Abstract

Hyperlipidemia is a powerful risk factor for coronary heart disease (CHD). It has been known for a long time that lipid-lowering drugs significantly reduce morbidity from CHD, thus proving a causal role for cholesterol in coronary events. Conversely, the relationship between low-density lipoprotein cholesterol (LDL-C) levels and stroke has been less clear and debated for many years. Recent data conclusively demonstrate not only the inverse epidemiological relationship of blood LDL-C with stroke, but also the efficacy of different strategies to attain cholesterol-lowering on stroke. They also dissipate lingering doubts about the possibility that lipid-lowering is linked to an increase in hemorrhagic stroke. However, despite current international lipid guidelines now strongly recommend aggressive lipid-lowering therapy in patients with atherosclerotic cardiovascular disease, including CHD and cerebrovascular disease (CeVD), secondary prevention patients are often undertreated with lipid-lowering therapies in routine clinical practice. This review highlights that patients with CHD and concomitant CeVD do not receive aggressive lipid-lowering therapy despite being at very high risk and with clear evidence of benefit from lowering LDL-C levels below current targets.

## 1. Introduction

Atherosclerotic cardiovascular disease currently accounts for the majority of deaths in most parts of the world [1]. Within the realm of cardiovascular disease, coronary heart disease (CHD) and stroke are the number 1 and number 3 causes of death, respectively.

Hyperlipidemia is a powerful risk factor for CHD [2]. It has been known for a long time that nonpharmacological as well as pharmacological cholesterol-lowering treatments significantly reduce morbidity from CHD, thus proving a causal role for cholesterol in coronary events. Conversely, the relationship between cholesterol levels and stroke has been less clear and debated for many years [3]. Anyhow, current international lipid guidelines now strongly recommend lipid-lowering therapy for secondary prevention in patients with atherosclerotic cardiovascular disease (ASCVD), including CHD and cerebrovascular disease (CeVD) [4,5].

Despite the overwhelming evidence that reducing low-density lipoprotein cholesterol (LDL-C) levels is highly beneficial in preventing recurrent ischemic events, secondary prevention patients are often undertreated with lipid-lowering therapies in community practice [6,7,8,9]. While appropriate statin therapy is an important goal in patients with CHD, it is still unclear whether CeVD patients are treated differently [7]. Furthermore, patients with CHD may have multisite artery disease involving additional vascular beds, including CeVD and/or peripheral artery disease (PAD), and a direct correlation exists between the number of arterial territories affected by atherosclerosis and the risk for future cardiovascular events [10,11]. Indeed, the presence of CeVD in patients with CHD has been found to be associated with increased long-term major adverse cardiovascular events (MACE). These are highest when the two conditions co-existed. Attainment of LDL-C treatment goals was related to lower risk for adverse events. Nevertheless, a large proportion of CHD patients with CeVD does not achieve lipid goals [7].

Recent guideline updates [4,5] specifically identified the presence of high-risk comorbidities as scenarios in which the addition of new lipid-lowering therapies should be considered if LDL-C levels remain high despite maximally tolerated statin therapy. Recent randomized controlled trials provide support for these recommendations by showing that, within a relatively short-time frame, these very high-risk patient groups have a lower risk of MACE with lower achieved LDL-C [12,13,14].

The present narrative review is aimed at updating the available evidence and the differences in cholesterol-lowering therapy in patients with isolated CHD vs. those with stroke and those with concomitant CHD and stroke, to stimulate a more aggressive approach in reducing LDL-C in all these very high-risk patients.

## 2. Lipid Lowering Therapy in Patients with Coronary Heart Disease

Treatment of derangements in lipid parameters is an essential objective for controlling the risk of cardiovascular complications in patients with established CHD. Numerous components of the lipid profile (LDL-C, triglyceride-rich lipoproteins (TGL), apolipoprotein B (apoB), lipoprotein (Lp) (a) and high-density lipoproteins (HDL)) have been identified as targets for therapy. Here, however, we will focus only on those with the largest body of evidence and with the widest acceptance in daily clinical practice.

### 2.1. Lowering LDL-Cholesterol: The Cornerstone of Lipid-Lowering Therapy

Since the pivotal Scandinavian Simvastatin Survival Study (4S) study in 1994 [15], many randomized controlled trials have tested the impact of LDL-C lowering interventions on the re-occurrence of ischemic events in patients with history of atherosclerotic CHD. A first summary of these results was reported in 2005 by the Cholesterol Treatment Trialists’ (CTT) Collaborators meta-analysis, where in patients with previous CHD a reduction of 1 mmol/L (38.7 mg/dL) of LDL-C was associated with a 21% relative risk (RR) reduction of recurrent events (RR: 0.79, 95% CI 0.76–0.81) [16].

The role of LDL-C reduction as the main driver of cardiovascular risk reduction in CHD patients has been further reinforced by the demonstration that a more pronounced decrease in LDL-C generates an incremental cardiovascular protection. A comprehensive quantification of the benefit of a more intensive vs. less intensive LDL-C lowering has been offered by a further meta-analysis of the CCT Collaborators [17] that included 170,000 patients participating into 26 randomized trials with statins. When in CHD patients less intensive were compared with more intensive regimens, a highly significant 21% further decline in the recurrence of major cardiovascular events was found per 1 mmol/dL (38.7 mg/dL) LDL-C difference at 1-year follow-up.

These consistent results represented the basis for recommending the use of high-intensity, high-dose statins in patients with previous CHD [5]. However, this pragmatic approach has been challenged by the observation that these patients even if treated with intense statin regimen remain at high risk if they do not achieve adequate levels of LDL-C [18]. This has opened the way to studies aimed at identifying the level of LDL-C (and the LDL-C lowering strategy) that is associated with the lowest risk of recurrence of ischemic events.

In the Improved Reduction of Outcomes: Vytorin Efficacy International Trial (IMPROVE-IT) [12], 18,144 patients who had been hospitalized for an acute coronary syndrome within the preceding 10 days and showing baseline LDL-C of 94 mg/dL were randomized to receive simvastatin (40 mg) or simvastatin (40 mg) plus ezetimibe (10 mg). The average LDL-C during the study was 53.7 mg/dL in the simvastatin plus ezetimibe group as compared with 69.5 mg/dL in the simvastatin-monotherapy group (P < 0.001). This difference translates into a 2% absolute risk reduction for a composite of cardiovascular death, nonfatal myocardial infarction, unstable angina requiring rehospitalization, coronary revascularization, or non-fatal stroke (RR: 0.936; 95% CI, 0.89–0.99; *p* = 0.016).

More recent trials with proprotein convertase subtilisin/kexin type 9 inhibitors (PCSK9i), have further evaluated whether cardiovascular protection in CHD patients extends down to very low levels of LDL-C. In the FOURIER study [19], involving 27,564 patients with atherosclerotic cardiovascular disease and LDL-C levels above 70 mg per deciliter when receiving statin therapy, 140 mg evolocumab administered subcutaneously every 2 weeks brought LDL-C levels down to 30 mg/dL (from a median baseline value of 92 mg/dL). Compared to placebo, patients receiving evolocumab after 2.2 years experienced 15% lower risk of the primary endpoint (a composite of cardiovascular death, myocardial infarction, stroke, hospitalization for unstable angina, or coronary revascularization) (RR: 0.85; 95% CI, 0.79–0.92; *p* < 0.001) and 20% lower risk of the secondary endpoint (a composite of cardiovascular death, myocardial infarction, or stroke) (RR 0.80; 95% CI, 0.73–0.88; *p* < 0.001). The ODYSSEY OUTCOMES trial obtained similar data in a population of 18,924 patients who had an acute coronary syndrome (ACS) 1 to 12 months earlier (3 months median interval) [20]. These patients, showing baseline LDL-C of 92 mg/dL while receiving maximally tolerated lipid-lowering therapies, were randomized to receive placebo or alirocumab 75 mg/dL every other week, with up-titrations to 150 mg/dL or down-titration to maintain an LDL-C between 50 and 25 mg/dL. After a 2.8-month follow, patients treated with alirocumab had a 15% reduction of a composite of death from CHD, nonfatal myocardial infarction, fatal or nonfatal ischemic stroke, or unstable angina requiring hospitalization) (RR: 0.85; 95% CI], 0.78–0.93; *p* < 0.001). In this study there also was a trend towards lower cardiovascular mortality and a significant reduction in total mortality (nominal *p* < 0.026).

A summary view of these studies is shown in Table 1. All these more recent findings were the foundation for the transition from the “high-intensity statin therapy” to the” high-intensity LDL-lowering therapy” concept [21], which has inspired the recommendations contained in the latest 2019 European Society of Cardiology (ESC)/European Atherosclerosis Society (EAS) guidelines [4]. These recommend an LDL-C goal of <55 mg/dL and an LDL-C reduction by at least 50% in patients at very high risk. This goal can be further lowered to 40 mg/dL in patients who experienced a second vascular event within 2 years while taking maximally tolerated statin-based therapy. Moreover, in the treatment algorithm these guidelines suggest to start with high-potency statins at the highest recommended/tolerable dose to reach the goal (Class I, Level A); if the LDL-goal is not achieved after 4–6 weeks, combination with ezetimibe is recommended (Class I, Level B); thereafter, if the LDL-C goal is not achieved after 4–6 weeks of combination therapy, the use of a PCSK9 inhibitor is recommended (Class I, Level B). If LDL-C is ≥40 mg/dL within 4–6 weeks and a recurrent ASCVD event occurs within 2 years, the addition of the PCSK9 inhibitor may be considered (Class IIb).

However, several aspects must be considered for the implementation of this stepwise LDL-lowering treatment in clinical practice. One is represented by the proportion of very-high risk patients who can attain the recommended LDL-C goal at each step. Using the nationwide SWEDEHEART register, which included 25,466 patients with ACS, it was simulated that the maximized use of high-intensity statins followed by addition of ezetimibe allowed the attainment of ESC/EAS LDL-C goal in about 48.3% [22]. Therefore 50.7% would still be eligible for PCSK9 inhibitors. The analysis of a contemporary, prospective Swiss cohort of 2521 patients hospitalized for ACS produced similar results [23]. All these data clearly indicate that the implementation of the ESC/EAS guidelines may favor the advent of a triple therapy for the appropriate management of LDL-C levels among patients at very high risk because of CHD [24].

Another crucial aspect is represented by the fact that the LDL-lowering stepwise approach may require time and a very efficient health care organization to be fully implemented in the routine clinical practice. This can create delays in bringing CHD patient to an adequate high-intensity LDL-lowering therapy. Therefore, it would perhaps be more recommendable that patients who have survived a coronary ischemic event receive maximal cholesterol-lowering therapy as soon as possible, at least combining high-potency statins with ezetimibe immediately after the event [25].

Finally, a still open question is how low we can go with LDL-C levels. Some information can be derived from the high intensity LDL-lowering trials [14,26]. Based upon such findings, we can assume that a detectable preventive benefit of LDL-lowering intervention in CHD patients can be found down to 20–30 mg/dL of LDL-C concentrations. Even among patients with LDL values below 10 mg/dL, this benefit was not associated with any increase of adverse events.

### 2.2. Targeting Triglyceride and Non-HDL-C: The Refinement of Lipid-Lowering Therapy

The abovementioned data fit well with the idea that LDL-C is an etiological factor of atherosclerosis [27]. However, an increasing number of evidences suggest that also TGL (very low-density lipoprotein or VLDL and their remnants) play an important role in atherogenesis [28]. This has been demonstrated to be true also in patients with established ASCVD [29] even if effectively treated with a statin [30]. Based on this and other findings it has been postulated the TG may significantly contribute to the so called “residual risk” thus representing a potential therapeutic target in secondary prevention [31].

The diagnosis HTG is complicated by the high heterogeneity and variability of TG measurements. In a joint statement of the EAS and the European Federation of Clinical Chemistry and Laboratory Medicine, it was indicated that non-fasting TG > 175 mg/dL should be considered as abnormal, while for fasting samples, abnormal concentrations correspond to TG > 150 mg/dL [32]. These values can be also taken as reference for patients with CHD. Moreover, it has been proposed that in HTG patients it may be useful to determine the non-HDL-C (calculated as total cholesterol minus HDL-C), which is a good estimate of all apoB-containing lipoproteins, e.g., VLDL and their remnants as well as LDL-C [4], and a powerful predictor of cardiovascular risk [31].

Two categories of medications are currently available in clinical practice to effectively target HTG: fibrates and marine-derived omega-3 fatty acids, such as eicosapentaenoic acid (EPA) and docosahexaenoic acid (DHA). Their estimated TG-lowering effect ranges between 30% to 50%. A review of pharmacological characteristics of these drugs is out the scope of this article. Our aim is to consider the available evidence of their benefit in reducing the risk of recurrence of ischemic events in patients with CAD. Table 2 summarizes these data.

When used in monotherapy, the protective effect of fibrates appears to be small, if any [33,34,35,36,37,38]. Their use in combination with statins did not apparently provide additional benefit [39,40], even though subgroup analyses indicate that patients with HTG and low HDL-C may be protected from such combination therapy [41]. According to previous reports and meta-analyses, the prescription of marine omega-3 fatty acids, mainly at a low dosage (1 g/day), in patients with CHD have generally failed to provide ASCVD protection [42,43]. More recently, however, the Reduction of Cardiovascular Events with Icosapent Ethyl–Intervention Trial (REDUCE-IT) tested whether a highly purified ethyl ester of EPA, icosapent ethyl, used at high dosage (4 g/day), determined a cardiovascular risk reduction in addition to protection afforded by statins [44]. REDUCE-IT enrolled 8179 patients (70.7% for secondary prevention of cardiovascular events) with baseline elevated TG (median 216 mg/dL) on a background statin therapy (median LDL-C 74 mg/dL). It was reported that icosapent ethyl, significantly reduced, by 25%, the risk of cardiovascular death, myocardial infarction, stroke, hospitalization for unstable angina, or coronary revascularization (RR: 0.75; 95% CI 0.68–0.83; *p* < 0.001). This benefit was even more pronounced (risk reduction 27%) in patients in secondary prevention. These interesting findings were further supported by the demonstration that, as compared with mineral oil, 4 g/day icosapent ethyl were able to significantly slow the progression of atherosclerotic plaques in 80 patients with documented coronary atherosclerosis and high TG (259 mg/dL) [45]. Unfortunately, another more recent study, the STRENGTH trial, failed to demonstrate any benefit of omega-3 in high-risk patients [46]. In this trial, 13,078 patients were randomized to receive a carboxylic acid formulation of EPA and DHA (4 g/day) or corn oil in addition to usual background therapies, including statins. Their baseline median TG level was 240 mg/dL and LDL-C was 75.0 mg/dL. The primary end point occurred in almost identical proportion of patients treated with omega-3 vs. corn oil (12% vs. 12.2%, respectively).

Therefore, even though it is reasonable to hypothesize that targeting TG (or non-HDL-C) in CHD patient is helpful for reducing their residual cardiovascular risk, the best way to do it remains controversial. To this regard, it must be reported that the 2019 ESC/EAS guidelines suggested that n-3 PUFAs (icosapent ethyl 2 × 2 g/day) should be considered in combination with a statin in high-risk patients (including CHD) showing TG levels between 135–499 mg/dL despite statin treatment (Class IIa, Level B); moreover, in high-risk patients who are at LDL-C goal but still present TG levels > 200 mg/dL, fenofibrate or bezafibrate may be considered in combination with statins (Class IIb, Level C).

## 3. The Role of Cholesterol-Lowering Therapy in Stroke

The role of hypercholesterolemia as a potent risk factor for CHD is currently undisputed, since a plethora of demonstrations that cholesterol-lowering interventions, both by drugs and non-pharmacological treatments, significantly reduce morbidity from coronary heart disease, thus closing the circle of the causality inference [2]. In a comprehensive meta-regression analysis, the use of statin and nonstatin therapies that act via upregulation of LDL receptor expression to reduce LDL-C were associated with similar relative risk (RRs) of major vascular events per change in LDL-C: lower achieved LDL-C levels were associated with lower rates of major coronary events independent of treatments used to attain the LDL reduction goal [47].

Conversely, the relationship between cholesterol levels and stroke has been much less clear. For long time, the tenet has been that hypertension is the main risk factor driving the risk of stroke, while the role of blood cholesterol was downplayed, due to the lack of demonstration of clear effects on stroke by drugs and treatments before the statin era [3]. Trials with statins have clearly shown decreased stroke incidence in treated populations [2,3]. This has for long fueled the concept that effects of statins on stroke are attributable to cholesterol-independent (neuro)protective properties of statins, “pleiotropism”, related to the interference by statins with the mevalonate pathway [48]. In more recent times this concept has been repeatedly challenged and the hypothesis refuted. The question was addressed a first time in a meta-regression of all cholesterol lowering interventions, where a significant relationship with total stroke was reported: the greater the magnitude of cholesterol-lowering with pre-statin drugs or interventions (such as ileal bypass), the greater the reduction of stroke, with no significant association with changes of HDL cholesterol levels, and inconsistent associations with reduction of TG [49].

The subsequent appearance of ezetimibe, a drug interfering with cholesterol absorption and not interfering with the mevalonate pathway, offered a second proof of concept [50]. Indeed, the IMPROVE IT trial [12], assessing the incremental cardiovascular benefit of adding the non-statin agent ezetimibe to statin therapy also documented a reduction in stroke, with the magnitude of stroke reduction fitting well the regression line from a previous analysis [50]. Most recent trials with PCSK9 inhibitors have made data available with other drugs with pure effects on LDL (and total) cholesterol, and with an unprecedented extent of LDL-C reduction. As of today, results from the FOURIER [19], ODYSSEY OUTCOMES [20] and SPIRE 1/2 [51] trials have allowed a further proof of concept, because the three drugs here used—evolocumab, alirocumab and bococizumab, respectively—reduce total and LDL-C by a mechanism totally independent of the mevalonate pathway and—this time—to a much more substantial extent compared with previous non-statin treatments. Indeed, all three trials perfectly fit the previously reported regression line of the relationship between the Delta of total cholesterol lowering and total stroke (total cholesterol here used because of the lack of cholesterol subfraction reporting of older trials; total stroke because of the uncertainty in the stroke etiology assessment in several trials examined in the meta-analysis and meta-regression [52].

When examined in their totality, these results (a) confirm that no special property of any cholesterol-lowering intervention has to be invoked to explain the reduction in stroke, fitting a log-linear relationship; (b) allow a precise estimate of the expected results on stroke in future intervention trials affecting total cholesterol. Specifically, the latest equation reported to predict the saving of strokes as a function of total cholesterol lowering is now: LnRR = −0.061 − 0.005 × (% cholesterol reduction). According to this equation, one should expect a RR of 0.851, 0.810 and 0.770 for a 20%, 30% and 40% reduction in total cholesterol, respectively.

These data conclusively demonstrate not only the inverse epidemiological relationship of blood cholesterol with stroke, but also the efficacy of cholesterol-lowering interventions, achieved with multiple strategies, on stroke. They also dissipate lingering doubts about the possibility that cholesterol lowering is linked to an increase in hemorrhagic stroke, which did not appear despite the extremely low levels of both LDL and total cholesterol achieved in the three above-mentioned PCSK9 inhibitor trials [19,20,51,53,54]. Favorable results apply to both primary and secondary stroke prevention, being consistent both in patients never having experienced a stroke before (and largely receiving it for other reasons), and in patients with previous stroke. In such patients, due to the higher absolute number of events, a higher absolute benefit can be expected, as recently shown [55]. Indeed, the Stroke Prevention by Aggressive Reduction in Cholesterol Levels (SPARCL) trial showed that the total events reduction by atorvastatin compared with placebo in 4731 participants with recent stroke or transient ischemic attack and no known coronary heart disease included 177 fewer cerebrovascular, 170 fewer coronary, and 43 fewer peripheral events. Over six years, an estimated 20 vascular events per 100 participants were avoided with atorvastatin treatment [55], providing a compelling argument for secondary prevention of stroke, where intensive lipid-lowering therapy prevented more than twice the number of first events.

These results obviously apply to both patients with and patients without co-existing CHD [56]. They also allow a prediction that the lower LDL and total cholesterol, the higher the expected benefit in terms of stroke reduction, as recently conclusively demonstrated [57]. There is apparent no token to pay for this, as the same trial has also shown that the incidence of intracranial hemorrhage and newly diagnosed diabetes did not differ significantly between the groups here randomized to a higher LDL-C target group (target range of 90 to 110 mg/dL—2.3 to 2.8 mmol/L) or to a lower target group (less than 70 mg/dL—1.8 mmol/L) [57]. Statins also appear to be effective in acute ischemic stroke [58].

Other lipid-related factors, beyond LDL cholesterol, appear to play a role in the risk of stroke. These include triglycerides and lipoprotein (a) (Lp(a)). In the REDUCE-IT, the reduction in triglycerides by Icosapent Ethyl, a pure preparation of the omega-3 fatty acid eicosapentaenoate, was associated with a significant 28% risk reduction of fatal and nonfatal stroke [44]. Data on new agents designed to lower plasma concentrations of Lp(a) are eagerly awaited.

## 4. The Role of Cholesterol-Lowering Therapy in Patients with Concomitant CHD and Prior Stroke

### 4.1. Attainment of Lipid Goals with Statins and Outcome in Patients with CHD and Concomitant CeVD

Until a few years ago, despite the well-established efficacy of lipid-lowering therapy, limited data were available on guideline attainment among high-risk coronary patients with preexisting cardiovascular disease, particularly among patients with stroke.

A Canadian study including ≈5000 high-risk ambulatory patients with CHD, CeVD, or both found that <25% met LDL-guideline–recommended targets [59]. In the Get-With-The-Guidelines-Stroke Registry, including 913,436 patients with an acute ischemic stroke or transient ischemic attack from April 2003 to September 2012, as many as 88,605 (9.7%) patients had concomitant CHD and CeVD [56]. Although 59.7% of patients with CeVD and CHD were receiving statins, only 62.3% and 28.5% met the LDL-C < 100 and the LDL-C < 70 mg/dL targets, respectively. Men were more likely to meet the LDL-C targets compared with women. Independent factors associated with attainment to the LDL guidelines among high-risk patients with preexisting stroke and CHD included older age, male sex, lack of vascular risk factors (except for atrial fibrillation), being on lipid-lowering agents before index event, and enrolment in the latest years.

However, it remained unclear whether the intensity of prescribed statin therapy differs for patients with CHD versus CeVD versus both CHD and CeVD. Recently, the Patient and Provider Assessment of Lipid Management (PALM) registry collected data on statin use, intensity, and core laboratory LDL-C levels for 3232 secondary prevention patients with CeVD only (*n* = 403), CHD only (*n* = 2202), and CHD and CeVD (*n* = 627) treated at 133 US clinics [7]. Fewer patients with CeVD only were received statin therapy (76.2% vs. 82.6% vs. 86.2%, *p* < 0.001) or treated at the guideline-recommended intensity (34.6%, vs. 49.8% vs. 50.4%, *p* < 0.001) than individuals with CHD and CeVD or those with CHD only. After risk adjustment, patients with CeVD only (adjusted OR 0.79, 95% CI 0.64–0.99) or CHD and CeVD (adjusted OR 0.73, 95% CI 0.61–0.87) were less likely to have an LDL-C < 100 mg/dL as compared with CHD only patients. Remarkably, there were no significant differences in the use of any statin therapy or guideline-recommended statin intensity between very high-risk individuals with concomitant CHD and CeVD and those with CHD only.

Even more recently, in a retrospective analysis of 10,297 CHD patients who underwent coronary revascularization categorized as having CHD alone or with multisite artery disease, 511 (5%) patients had CHD + CeVD and 417 (4%) had CHD + CeVD + PAD, respectively [60]. Attainment of LDL-C treatment goals was related to lower risk for adverse events. Regrettably, less than half of the patients had attained LDL-C < 70 mg/dL, and less than a quarter <55 mg/dL, regardless of the vascular site involved. These findings remain a source of concern, in the light of recent European guidelines [4] advising further reduction of LDL-C target levels to less than 55 mg/dL in patients with ASCVD, and even under 40 mg/dL in those with recurrent events, which is more expected to occur in CHD patients with multisite artery disease.

### 4.2. American vs. European Guidelines for the Management of LDL-C in These Very-High Risk Patients

Guidelines for the management of blood lipids were very recently updated in the United States and Europe [4,5].

The guidelines have a lot of similarities. Both recommend to markedly lower LDL-C as a significant modifiable risk factor and consider using non-statin agents, such as ezetimibe and PCSK9 inhibitors, in addition to lifestyle changes and high-intensity statins for a further reduction of LDL-C levels. At the same time, the guidelines have several relevant differences, including the concepts of treatment thresholds (American) vs. treatment goals (European) and the specific classes for recommendation, chiefly in secondary prevention [61] Table 3.

Indeed, in the European guidelines all patients with an ACS as well as patients with established ASCVD, including patients with stable angina, previous documented coronary revascularization, stroke, transient ischemic attack (TIA), or PAD are classified as very high risk, whereas in the American guidelines a patient with ACS must also have multiple high-risk features or more than one previous ASCVD event.

In secondary prevention of patients at very high-risk for ASCVD, the American guidelines recommend high-intensity statin to achieve an LDL-C reduction of ≥50%. If LDL-C levels remain above the threshold of ≥70 mg/dL despite maximally tolerated statin therapy, the guidelines recommend adding a non-statin agent. However, in secondary prevention of very high-risk patients, an LDL-C reduction of ≥50% from baseline and an LDL-C goal <55 mg/dL are both recommended in the European guidelines (class I, level A). Moreover, in patients with ASCVD who experienced a second vascular event within 2 years while taking maximally statin therapy an LDL-C goal of <40 mg/dL may be considered (class IIb, level B).

Finally, for very high-risk patients the European guidelines recommend the addition of non-statin agents to high-intensity maximal statin in a sequential way (ezetimibe first, then PCSK9 inhibitors; both are class I recommendations) if the LDL-C is >55 mg/L. In contrast, the American guidelines recommend the addition of ezetimibe (class I) to high-intensity statin if the addition of PCSK9 inhibitors is considered. This is considered reasonable if the LDL-C is above the threshold of ≥70 mg/dL (class IIa).

### 4.3. The Role of New Lipid-Lowering Therapies

The efficacy of the addition of ezetimibe to simvastatin for the prevention of stroke and other adverse cardiovascular events after ACS was investigated in IMPROVE-IT, with a focus on patients with a stroke before randomization [62]. Patients with a history of stroke at baseline were more likely to be already treated with lipid-lowering therapy prior to the index ACS event (58% vs. 35%; *p* < 0.001) and had lower LDL-C levels at the time of the index ACS event (87 mg/dL vs. 95 mg/dL; *p* < 0.001). Despite this evidence, achieved LDL-C values were similar at 1 year across subgroups (50–51 mg/dL with ezetimibe/simvastatin and 67–68 mg/dL with simvastatin alone), which means a consistent 17 mg/dL between treatment difference in LDL-C within each subgroup. Patients with a history of prior stroke were at high risk for recurrent stroke after randomization, including ischemic and hemorrhagic forms, and they also had a greater risk of MI, death, and the primary trial end point. A significant reduction in both first and total (first and subsequent) ischemic strokes, with a non-significant increase in hemorrhagic stroke in patients in whom ezetimibe was added to background statin therapy was observed. More specifically, patients with a prior stroke demonstrated an absolute risk reduction of 8.6% for stroke of any etiology (10.2% vs. 18.8%; HR, 0.60; 95% CI, 0.38–0.95; *p* = 0.030) and 7.6% for ischemic stroke (8.7% vs. 16.3%; HR, 0.52; 95% CI, 0.31–0.86; *p* = 0.011) with ezetimibe added to simvastatin therapy. For the first time, as pointed-out above, the consistency of these findings supported the benefit of LDL-C lowering on stroke prevention through a non-statin mechanism.

Subsequently, a pre-specified analysis from ODYSSEY OUTCOMES determined whether polyvascular disease, including CeVD and PAD, influenced risks of MACEs and death and their modification by alirocumab in patients with recent ACS and dyslipidemia despite intensive statin therapy [13]. Of 18,924 randomized patients, of whom 9462 were assigned to the alirocumab group and 9462 to the placebo group, 795 (4.2%) had concomitant CeVD and 149 (0.8%) had polyvascular disease in all 3 vascular beds. At baseline, median LDL-C (quartile 1, quartile 3) was higher in patients with polyvascular disease, with values of 86 mg/dL (73, 103 mg/dL) in patients with a single-district vascular disease, 90 mg/dL (75, 109 mg/dL) in CHD and CeVD, and 95 mg/dL (80, 115 mg/dL) in polyvascular disease in 3 beds (*p* < 0.0001). Overall, in the ODYSSEY OUTCOMES trial the incidences of MACE in the placebo and alirocumab groups were 11.1% and 9.5%, respectively, with a corresponding absolute RR of 1.6% (95% CI: 0.7% to 2.4%; *p* = 0.0003).

This overall efficacy reflects a gradient of absolute risk and absolute RR according to the number of diseased vascular beds. For patients in the placebo group the incidence of MACEs was 10.0% in isolated CHD, 21.1% in CAD and CeVD and 39.7% in those with CHD, CeVD and PAD. The corresponding absolute RR with alirocumab was 1.4% (95% CI: 0.6% to 2.3%) in isolated CHD, 2.7% (95% CI: −2.9% to 8.2%) in CHD and CeVD, and 13.0% (95% CI: −2.0% to 28.0%) in those with CHD, CeVD and PAD (interaction *p* = 0.0006). For all-cause death, similar to MACEs, there was a gradient of absolute risk and of absolute RR with alirocumab. Thus, patients with ACS and concomitant CeVD and dyslipidemia despite intensive statin therapy gain relevant absolute benefit from PCSK9 inhibition with alirocumab.

In the FOURIER trial a prespecified analysis of cerebrovascular events in the overall trial population and in patients stratified by prior stroke was performed [54]. In particular, the efficacy of evolocumab to reduce overall stroke and stroke subtypes, as well as the primary cardiovascular composite end point by subgroups according to a history of stroke was done. Evolocumab significantly reduced all stroke (1.5% vs. 1.9%; *p* = 0.01) and ischemic stroke (1.2% versus 1.6%; *p* = 0.005), with no difference in hemorrhagic stroke (0.21% vs. 0.18%; *p* = 0.59). These findings were consistent across subgroups, including among the 5,337 patients (19%) with prior ischemic stroke in whom the hazard ratios (95% CIs) were 0.85 (0.72–1.00) for the cardiovascular composite, 0.90 (0.68–1.19) for all stroke, and 0.92 (0.68–1.25) for ischemic stroke (*p* interactions, 0.91, 0.22, and 0.09, respectively, compared with patients without a prior ischemic stroke).

These findings clearly indicate that patients with prior CeVD and additional atherosclerotic risk factors benefit from lowering LDL-C levels below current targets.

## 5. Conclusions

A large body of evidence derived from randomized controlled trials emphasizes the importance of intensive, sustained LDL-C reduction in CHD patients. The inverse epidemiological relationship of LDL-C with stroke and the efficacy of cholesterol-lowering interventions, achieved with multiple strategies, on stroke has also been recently demonstrated in both primary and secondary stroke prevention. Furthermore, doubts about the possibility that cholesterol lowering is linked to an increase in hemorrhagic stroke have been dissipated. However, despite this overwhelming evidence, secondary prevention patients are often undertreated with lipid-lowering therapies in community practice. From this review it emerges that both the implementation of early intense treatment and the achievement of LDL-C target are largely suboptimal in the absence of concomitant CHD, and that patients with CHD and concomitant CeVD do not receive aggressive lipid-lowering therapy in spite of clear evidence of benefit from lowering LDL-C levels below current targets.

## Figures and Tables

**Table 1 jcm-10-00886-t001:** Randomized cardiovascular outcomes study with high intensity LDL-lowering therapy in patients with coronary artery disease.

Trial	Mean Reduction in LDL Cholesterol; mmol/L (mg/dL)	Outcome	RR (95% CI) (per mmol/L)
CTT meta-analysis (high-intensity vs. standard statin; subgroup < 2.0 mmol/L) [17]	1.71 (66) vs. 1.32 (50)	MI, CHD death, stroke, coronary revascularisation	0.71 (0.56–0.91)
IMPROVE-IT (ezetimibe plus simvastain vs. simvastatin) [12]	1.55 (70) vs. 1.40 (54)	CV death, MI, stroke, UA, coronary revascularisation	0.94 (0.89–0.99)
FOURIER (evolocumab plus high-dose statin ± ezetimibe vs. high-dose statin ± ezetimibe) [19]	2.37 (92) vs. 0.78 (30)	CV death, MI, stroke, UA, coronary revascularisation	0.85 (0.79–0.92)
ODYSSEY OUTCOMES (alirocumab plus high-dose statin ± ezetimibe vs. high-dose statin ± ezetimibe) [20]	2.37 (92) vs. 1.37 (53)	MI, CHD death, stroke, UA	0.85 (0.78–0.93)

CHD, coronary heart disease; CV, cardiovascular; MI, myocardial infarction; UA, unstable angina.

**Table 2 jcm-10-00886-t002:** Randomized cardiovascular outcomes study with triglycerides-lowering therapy in patients with coronary artery disease.

Trial	Population	Treatment	Primary Endpoints	Follow-Up	TG Reduction In all Partecipants	Risk Reduction for Partecipants with Previous CVD	Risk Reduction in Subgroup with Baseline HTG and/or Low HDL
**Fibrates**							
CDP (1975) [34]	3892 patients with CHD	Clofibrate 1.8 g/day vs. placebo	Non fatal MI+ CHD death	5.0 years	22.3% (for clofibrate)	38%	38%
VA-HIT (1999) [35]	2531 patients with low HDL-C and CHD	Gemfibrozil 1200 mg/day vs. placebo	Non fatal MI + CHD death	5.1 years	31%	27%	28%
BIP (2000) [36]	3090 patients with previous MI or angina	Bezafibrate 400 m/day vs. placebo	MI+ sudden death	6.2 years	21%	7.3%	39.5%
FIELD (2005) [37]	2131 patients with DMT2 and CVD	Fenofibrate 200 mg/day vs. placebo	non fatal MI + CHD death	5 years	21.9%	+ 2.0%	27.0%
ACCORD (2010) [40]	2016 patients with DMT2 and CVD	Fenofibrate 160 mg/day + simvastatin 40 mg/day vs. simvastatin 40 mg/day	MI + stroke+ CV death	4.7 years	22.0%	10.0%	28%
ACCORDION (2017) [41]	1620 patients with DMT2 and CVD	Fenofibrate 160 mg/day + simvastatin 40 mg/day vs. simvastatin 40 mg/day	MI+ stroke+ CV death	9.7 years	14.4%	7%	27%
**High dose omega-3 fatty acids**							
REDUCE-IT (2019) [45]	5875 patients with CVD	EPA 4g vs. mineral oil	MI + stroke + coronary rivascularization-unstable angina + CV death	4.9 years	19.7%	27%	38%
STRENGHT (2020) [47]	7316 patients with CVD	EPA+ DHA 4 g vs. corn oil	MI + stroke + coronary rivascularization-unstable angina + CV death	3.5 years	19.9%	6%	1%

CDP, Coronary Drug Project; DMT2, diabetes mellitus; CVD, cardiovascular disease; CHD coronary heart disease; MI, myocardial infarction; CV, cardiovascular; HTG, hypertrgliceridemia; HDL, high-density lipoprotein.

**Table 3 jcm-10-00886-t003:** Comparison of LDL-C lowering strategies in CAD and CVD patients as recommended by the American and European Guidelines.

	American Guidelines	European Guidelines
**CHD patients**		
Risk stratification	Patients with multiple major ASCVD events * OR multiple high-risk conditions ^§^ must be considered at very-high risk	All patients with CAD must be considered at very-high risk
Treatment strategies	Initiate with high intensity or maximal statin therapy to lower LDL-C levels by ≥50% (Class I, Levels A)	Initiate with high-dose, high intensity statin therapy (Class I, Level A)
	If on maximal stain therapy LDL-C > 70 mg/dL add ezetimibe (Class IIa); If iPCSK9 is considered, add ezetimibe to maximal statin therapy before adding iPCSK9 (Class I, Levels B)	Revaluation after 4–6 weeks to determine whether a reduction of >50% from baseline and LDL-C goal < 55 mg/dL have been reached (Class I, Level A) (LDL-C goal < 40 mg/dL may be considered in patients with recurrent events (Class IIb, Level B))
	If on clinically judged maximal LDL-C lowering therapy and LDL-C > 70 mg/dL adding iPCSK9 is reasonable (Class IIa, Level A)	If not consider adding ezetimibe and eventually iPCSK9 (Class I, Levels B)
**CeVD patients**		
Risk stratification	Patients with multiple ASCVD events OR multiple high-risk conditions must be considered at very-high risk	All patients with ischemic stroke or TIA (particularly if recurrent) must be considered at very-high risk
Treatment strategies	Initiate with high intensity or maximal statin therapy to lower LDL-C levels by ≥50% (Class I, Levels A)	Initiate with high dose statin therapy (Class I, Level A)
	If on maximal stain therapy LDL-C > 70 mg/dL add ezetimibe (Class IIa); If iPCSK9 is considered, add ezetimibe to maximal statin therapy before adding iPCSK9 (Class I, Levels B)	Revaluation safter 4–6 weeks to determine whether a reduction of >50% from baseline and LDL-C goal < 55 mg/dL have been reached (LDL-C goal < 40 mg/dL in patients with recurrent events (Class I, Level A)
	If on clinically judged maximal LDL-C lowering therapy and LDL-C > 70 mg/dL adding iPCSK9 is reasonable (Class IIa, Level A)	If not consider adding ezetimibe and eventually iPCSK9 (Class I, Levels B)

* Multiple ASCVD events include MI, stroke and PAD; ^§^ Multiple high-risk conditions include: age > 65 years, HeFH, DM, HTN, CKD, smoking, LDL-C > 100 despite lipid lowering therapy, CHF. CAD, coronary artery disease; CeVD, cerebrovascular disease; ASCVD, atherosclerotic cardiovascular disease; LDL-C, low-density lipoprotein cholesterol; iPCSK9, inhibitors of PCSK9, MI, myocardial infarction; PAD, peripheral artery disease; HeFH, heterozygous familial hypercholesterolemia; DM, diabetes mellitus; HTN, hypertension; CKD, chronic kidney disease; CHF, congestive heart failure.

## Data Availability

Data is contained within the article.

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
