# Peer review of "Lipid-Lowering Therapy in Patients with Coronary Heart Disease and Prior Stroke: Mission Impossible?"

_jcm, 2021, doi:10.3390/jcm10040886_

Round 1

Reviewer 1 Report

The author did an extensive literature review, focusing on the benefit of lipid lowering therapy in patient with cardiovascular disease, and highlighting on the high-risk population who are still undertreated compared to other CV disease such as coronary artery disease.

stroke was not considered a vascular risk factor and patients with ischemic strokes were managed according to primary prevention guidelines as recommended by ASA in 2006. SPARCL trial was a landmark trial that changed the management of these high-risk patients.

-I will suggest that the author provide in the review data from the SPARCL trial that addressed among few dyslipidemia managements in stroke patients.

-Another suggesting is to focus on the role of statin in acute stroke and refer to this metanalysis by Hong KS et al, Statins in acute ischemic stroke: a systematic review.

-Another statement the author can use it to emphasize on the net benefit of acute statin therapy in patient with cerebrovascular disease which outweighs the concerns of hemorrhagic stroke.

-The author can also refer to this meta-regression analysis by Silverman et al, Association between lowering LDL-C and cardiovascular risk reduction among different therapeutic interventions: a systematic review and meta-analysis, published in JAMA in 2016 that compared the effect of statin versus non-statin lipid lowering therapy on the rate of vascular events.

-Another topic that the author can address and provide data on previous studies, is to mention the incidence of stroke in patient with elevated TG level, and what is the current approach to treat these patients.

Reviewer 2 Report

Well done review of literature for lipid lowering therapy in CAD and cerebrovascular disease.

Correctly shown that the CeVD patients are usually less intensely treated.

No major issues - would suggest making a table showing different American and European guidelines for CAD and CeVD lipid lowering respectively.

Reviewer 3 Report

This work is focused on lipid-lowering therapy in the setting of CVD, in particular coronary heart disease and stroke. The most recent evidences in this field, including American and European guidelines, have been reviewed showing the importance of an intensive LDL-C reduction to improve cardiovascular outcome. However, the achievement of LDL-C target is still suboptimal even in patients at very high risk for a CV event. The paper is clear, well organized and well written. I have no comment for the authors.
